# Development of a Three-Dimensional Nerve Stretch Growth Device towards an Implantable Neural Interface

**DOI:** 10.3390/mi13101558

**Published:** 2022-09-20

**Authors:** Xiao Li, Yongguang Chen, Xikai Tu, Hailong Huang

**Affiliations:** 1School of Mechanical Engineering, Hubei University of Technology, Wuhan 430068, China; 2Department of Rehabilitation Medicine, Zhongnan Hospital of Wuhan University, Wuhan 430077, China

**Keywords:** nerve stretch growth, neural interface, three dimensional, motor control, axon growth

## Abstract

Because of rising traumatic accidents and diseases, the number of patients suffering from nerve injury is increasing. Without effective rehabilitation therapy, the patients will get motor or sensory function losses or even a lifelong disability. As for amputees, neural interface technology can be used to splice nerves and electrical wires together in a way that allows them to control an artificial limb as if it was a natural extension of the body. However, the means the need for an autologous nerve to stimulate axonal regeneration and extension into target tissues, which are limited by the supply of donor nerves. Based on the principle of mechanical force regulating axon growth, in this paper, we developed a three-dimensional nerve stretch growth device for an implantable neural interface. The device consists of three motors controlled by single chip microcomputer and some mechanical parts. The stability and reliability of the device were tested. Then, we used neurons derived from human pluripotent stem cells by small chemical molecules to explore the optimal three-dimensional stretch culture parameters. Furthermore, we found that the axons were intact through 10 rotations per day and 1 mm of horizontal pulling per day. The results of this research will provide convenience for patients treated through an implantable neural interface.

## 1. Introduction

Nerve injury is a very common and serious clinical trauma, which may lead to partial or complete loss of motor, sensory, and autonomic nerve functions [1]. Every year, more than 5 million peripheral nerve repair operations are performed by surgeons worldwide [2]. There are a large number of amputees in these patients with peripheral nerve injury. For amputated patients, the limbs with neural connections are cut off, and their motor and sensory functions can be reconstructed by installing intelligent prostheses. The information interaction between human neural signals and intelligent prostheses can be realized through neural interface technology [3].

A neural interface is to build a bridge between the human nervous system and external electronic devices (intelligent prosthesis) for data information interaction, extracting motor or sensory nerve signals from the end of the amputee limb, and decoding the amputee’s active consciousness nerve signals through neural interface technology to control the movement of the prosthesis. Electromyography [4], EEG [5], the cerebral cortex [6], and peripheral nerves [7] can be used as the signal source of prosthetic control instructions. Among them, the resolution of the cerebral cortex and peripheral nerve signals is the highest. Theoretically, depending on high-quality nerve signals, amputees can manipulate prosthetics with multiple degrees of freedom as easily as their own limbs. However, this method requires the electrode to be directly inserted into the patient’s cerebral cortex or peripheral nerve to collect nerve signals. As a foreign body, it will not only damage the normal nerve, but also easily cause the immune response of organisms, form scar tissue [8] around the electrode, and interfere with the extractions of signals.

The development of highly biocompatible electrodes that can be implanted into organisms for a long time is the key to the success of neural interfaces [8]. Using biological tissue engineering technology, a segment of a nerve bundle is cultivated in vitro to connect the host nerve cells and electronic equipment. On the one hand, it can induce the host nerve to actively grow and connect to the nerve bundle cultivated in vitro. On the other hand, it can also avoid the damage of electronic equipment to the host nerve, which will be of great significance to the application of the nerve interface [9]. In recent years, researchers have made great progress in biocompatible electrodes by optimizing the nerve electrode tissue interface, evaluating the electrochemical performance of devices, the recording performance of biological signals, and the stability of the nerve interface [10]. Tsuyoshi et al. [11] used multi-walled carbon nanotube dispersions and hydrogels to produce a biocompatible and highly conductive gel composite, which was implanted into living tissue to collect the electrocardiography of uneven heart tissue. Park et al. [12] studied the performance of an encapsulated nerve interface composed of thermal epoxy resin and polyethylene terephthalate film deposition, and they confirmed that body fluid would not produce an immune response to the deposition of paling film on the packaging surface in the body for up to 18 months.

Although the use of a neural interface based on tissue engineering can solve the problem of electrode compatibility, the separation and decoding of neural signals are still difficult due to the complexity of neural connections. In order to make the length of the nerves cultured in vitro controllable and arranged regularly, a new axon stretch growth technology has entered people’s vision. In the experiment of axonal stretch tolerance, Pfister et al. [13] found that by gradually increasing the axonal stretch speed, the nerve bundle formed by rat dorsal root ganglion cells could grow up to 10 cm of nerve tissue in less than two weeks. Scanning electron microscopy and cellular immunohistochemistry showed that the axon morphology was complete, and the calcium activity was normal. Kameswaran et al. [14] proposed that the bottleneck of current neural interface technology can be solved by combining neural stretch culture technology with neural interface technology. After embedding electrodes on the stretch membrane, the stretch nerve bundle can be encapsulated in a nerve catheter with agarose. When transplanted into the body, one end of the stretch nerve bundle is connected with the host nerve, and the other end is connected with electronic equipment, so it can be used to control the prosthetic limb. In order to better separate neural signals, Katiyar et al. [15] separately cultured motor neurons and sensory neurons and also obtained functional long nerve bundles.

However, these studies were carried out under two-dimensional (2D) culture conditions, as shown in Figure 1. First, the membranes on both sides are fixed on the substrate respectively, and the width of the overlapping area between the two membranes is guaranteed to be 3–5 mm. Then, the suspension of nerve cells is dropped on the junction of bilateral membranes and cultured for 5–10 days until stable synaptic connections are formed between nerve cells. Finally, one slowly moves the towing rod, which drives the top membrane to move, so as to pull the nerve axon toward the top membrane. When the membrane movement speed is close to the growth speed of nerve axons, it will not cause nerve rupture but will cultivate a section of regularly arranged nerve tissue. Cells grown in this way are usually two-dimensional. In fact, the nerve bundle in an organism is a three-dimensional cylinder. Only the three-dimensional shape of the nerve bundle can be more directly applied to patients. Therefore, it is necessary to construct a three-dimensional cell culture system to verify that the mechanical stretching stimulation is still effective for axonal directional growth.

In this study, according to the principle of regulating axon growth by mechanical force and based on the three-dimensional structure of human nerve cells, we have developed a set of a three-dimensional axon stretch cultivation device for rapidly cultivating regular nerve tissue. The device consists of three motors controlled by a single-chip microcomputer and some mechanical parts, and it can realize rotation and straight-line movement. The circular basal culture base is used as the cell carrier, and the uniform rotation of the culture base ensures that the cells adhere to the films uniformly, and the straight-line movement realizes the axon stretching. We have carried out a displacement accuracy test and angular velocity accuracy test on the device respectively. More than 95% of the data measured by the device under the single step test of 1 μm were 1.0012 ± 0.166 μm, which basically met the error accuracy requirements, and more than 90% of the relative errors were less than 2%. In the test data of the minimum speed of 0.5 rpm, more than 95% of the data were 0.50 ± 0.03 rpm, and there was no gross error, which met the accuracy requirements of the experiment. The results of two test experiments showed that the three-dimensional axon stretching growth device can be used for an axon-stretching growth culture of nerve cells. Then, we used neurons derived from human pluripotent stem cells by small chemical molecules to explore the optimal three-dimensional stretch culture parameters. Furthermore, we found that the axons were intact through 10 rotations per day and 1 mm of horizontal pulling per day. The results of this research will provide convenience for patients treated through an implantable neural interface.

## 2. Materials and Methods

### 2.1. Design of a Three-Dimensional Neural Cell Stretch Growth Device

The three-dimensional stretch culture of neural cells is equivalent to the simultaneous stretch culture of multiple two-dimensional neural cells. The schematic diagram of a three-dimensional stretch culture device of neural cells is shown in Figure 2. The length, width and height of the ABS base were 540 × 320 × 30 mm, the cell culture tank was 250 × 170 × 40 mm, the maximum stretch distance of the pull rod was 50 mm, the diameter of the membrane attached to rotating carrier and stretching carrier was 10 mm, and the thickness of the membrane was 50 μm. Two closely connected transparent films were stuck on the towing rod in a circular shape, and the neural cell suspension was dropped on the circular basement membrane. After several days of slow rotation culture, the neural cells were evenly attached to the basement membrane, and then the horizontal stretch culture was carried out like a two-dimensional axonal stretch culture [16].

The equipment consisted of two parts: the mechanical system and control system (Figure 3 and Figure 4). The mechanical system was divided into three parts: the horizontal neural cell stretch culture subsystem, culture pedestal rotation subsystem, and pedestal rotation subsystem. Each subsystem had a mutually independent motor as the driving source. Under the joint action of the three subsystems, the cell culture tank completed the stretching and rotation culture of nerve cells.

As is shown in Figure 4, the three-dimensional nerve stretch growth device included a cell culture chamber (support block, support rod, towing block, towing rod, cell culture vehicle, transparent lid), connection block, liner displacement table, coupling, and three motors, etc. Among them, the cell culture vehicle in the cell culture chamber was made of polyether ether ketone (PEEK). The support block, towing block, and connection block were made of polytetrafluoroethylene (PTFE). The transparent lid was made of transparent polyester carbonate (PC), and other supporting rods, towing rods, and other mechanical components were made of 304 stainless steels. These materials have good biocompatibility, excellent corrosion resistance, and can withstand high-pressure steam sterilization. At different stages of axon stretch growth, the three motors received control commands to carry out forward rotation, reverse rotation, start, stop, reset, acceleration and deceleration, and other movements. The three motors coordinated with each other to make axons grow in a given direction. Motor A could drive the liner displacement table to move horizontally. The slider on the liner displacement table was fixedly connected with the stretch connecting block, which drove the movement of the towing rod and the towing block. The towing block drove one end of the cell culture base to move, and the stretch membrane attached to the culture base also moved synchronously, so as to pull the neural cells attached to the stretch membrane. Motor B was used to control the rotation of the substrate to ensure that the cells were evenly attached to the circular membrane under the action of gravity. Motor C could rotate the whole culture system at a certain angle to promote the growth of axons in one direction. The bottom of the membrane was inlaid with electrode contacts, and the bottom was connected with wires, which could be connected with external electronic equipment for signal recording and electrical stimulation of neurons.

This system used one electric linear displacement table and two Maxon motor kits as the driving devices. The electric linear displacement table was composed of a stepping motor and a linear slide. The entire stroke of the slide was 50 mm, and the helical pitch was 1 mm. The stepping motor was a two-phase stepping motor with a stepping angle of 1.8°. The Maxon motor kit consisted of a planetary gearbox, an EC 45 flat Ø 42.9 mm brushless DC motor, and a mile encoder. Among them, the reduction ratio of the planetary gear box was 126:1. The rated speed of the EC 45 brushless DC motor was 3070 rpm, and the rated torque was 0.0569 Nm. The number of pulses per revolution of the mile encoder was 2048, with two-phase channels A and B. The main control chip of the control system was a 32-bit high-performance STM32F407 microcontroller, whose core was ARM Cortex-M4, and the maximum clock frequency could reach 168 MHz. This system used an EBF-MSD4805 driver to drive the stepping motor and Copley Controls-ACJ-055-18 driver to drive the Maxon servo motor, respectively.

The three-dimensional nerve pulling cultivation equipment could realize rotation and pulling at the same time. The two shafts in the culture tank cooperated with each other to ensure that the two shafts could move synchronously and transmit torque. One shaft was designed with a rectangular groove, and the other shaft was matched with it, as shown in Figure 4. It could not only realize the synchronous rotation movement of two shafts with equal torque, but also meet the horizontal linear displacement movement. When it needed to rotate, the motor B drove the gear shaft to rotate through the coupling, so as to rotate the cell culture base, keep the nerve cells under uniform stress, and enable the nerve cells to adhere to the films evenly. When it needed to be stretched, the motor A drove the ball screw and the liner displacement table rail to generate linear motion, thus driving the connection block to move and then driving the towing rod to move. Finally, the axon could be rotated and stretched at the same time.

### 2.2. Micro-Displacement and Angular Velocity Measurement

If neurons are subjected to excessive force or speed, their axons will break, or the cell body will fall off the membrane. Before the cell culture, in order to ensure the stability of the nerve cell stretch growth process, displacement accuracy test and angular velocity accuracy test experiments needed to be carried out on the three-dimensional axon stretch device. The former was to prevent axon fracture caused by a too-large displacement step; the latter was to prevent the nerve cells attached to the membrane from falling off due to excessive rotation speed. An XL-80 laser interferometer of the Renishaw company in Britain was used to measure and record the displacement data of the stretch unit of the three-dimensional axon stretch device, and the actual speed data of the motor were collected by the MILE encoder of Maxon motor.

As for the horizontal displacement test experiment, according to the previous research of our team, the most suitable pulling displacement speed of nerve cells was about 1 mm every day, and the step distance was 1 μm [16]. Therefore, in this experiment, we directly set the single step displacement of the stepping motor to 1 μm, and the time interval between each two steps was 6 s, so as to ensure that nerve cells had enough time to adapt to growth. For each 1 μm of the stepping motor, the laser interferometer would automatically record the current displacement value, recording it for 500 times and repeating the above experimental process 10 times.

As for the angular speed test experiment, according to the centrifugal stress analysis [17], the minimum rotation speed of the culture base in our experiment was 5 r/d (Table 1). The Maxon motor kit used in this system was equipped with planetary gear reducer, and the maximum reduction ratio was 126. After conversion, the actual minimum speed of the EC 45 motor was 0.5 rpm. After debugging and testing the motor, we set the speed of the motor to 0.5 rpm, and the actual speed of the motor was fed back to the upper computer in real time. The data were sampled every 5 s, a total of 500 times, and the above process was repeated 10 times.

### 2.3. Cell Source and Culture

The cell used in the study was V2a excitatory glutamatergic neurons, which was obtained according to the induced methods [18] described by Butts J.C. In short, human pluripotent stem cells cultured on the MEF were differentiated into neuroepithelia, firstly under the induction of SB431542 (2 μM), DMH1 (2 μM), and CHIR99021 (3 μM) for one week. Then, the neuroepithelia were directly differentiated into spinal ventral neural progenitors with the induction of retinoic acid (RA, 0.1 μm), purmorphamine (pur, 0.1 μm), and DAPT (1 μm) for 7–15 days. The neural progenitor cells were further matured to V2a neurons in neural maturation medium supplemented with 10 ng/mL of BDNF, GDNF, CNTF, and IGF. The cell immunofluorescence analysis was performed using the TuJ-1 antibodies.

### 2.4. Optimization of Cell Rotation Speed

A too-fast rotation speed will cause neurons to fall off or an uneven distribution of neurons on the circular membrane. In order to find a suitable rotation speed, we cultured neurons at different rotation speeds (see Table 1) and statistically analyzed the number of neurons on the membrane. The diameter of the barrel membrane was about 1 cm. Briefly, the derived neurons were placed within 5 mm of the circular basement membrane lit by a micro-pipette, where the membrane was cut for the stretching culture. After the culture base was rotated 90 degrees, an equal amount of cell suspension was dropped, and then, it was left standing for 1 h and repeated 4 times until cells were distributed on the circular membrane. Then, after 5 days of rotation cultivation at the rotation speed (see Table 1), the number of cells on the circumference of the cell culture base was observed with an electron optical microscope. In order to facilitate observation and statistics, we divided the circumference of the cell culture seat into six sectors, and we observed the number of cells in the corresponding sector after each sector moved at different rotational speeds. The experiment was repeated five times at each speed, and the number of cells in each sector was analyzed by an ANOVA.

### 2.5. Neural Cells Stretch Growth

Before culture, the films were treated with PDL followed by rat-tail collagen. Then, the neurons were plated within 5 mm of the circular membrane lit by a micro-pipette. After each 90° rotation, the cells were placed once until the cells were distributed over the entire annular membrane. Cultures were maintained in the neural maturation medium supplemented with 10 ng/mL of BDNF, GDNF, CNTF, and IGF. At the same time, the film, rotated at a constant speed of 10 revolutions per day, allowed 10 days for axons to span the dividing region between the gap of the sheared film. Axon stretch growth began at a rate of 1 mm/day, with 1 μm steps every 86.4 s over 1000 iterations.

## 3. Results

### 3.1. Displacement Measurement

As shown in Figure 5, the probability density fitting distribution curve of 500 measurement points with a single step of 1 μm was recorded 10 times, repeatedly. It can be seen from the histogram that more than 95% of the 5000 single-step displacement data were 1.0012 ± 0.166 μm. After averaging, the overall fitting curve with less variance fluctuation was obtained, which approximately conformed to the normal distribution. It can be seen from Table 2 that the average value of the displacement data measured in each experiment was 1.0012 ± 0.083 μm. The average values of each single-step one-μm experiment tended to be consistent, with little fluctuation, and there was no significant difference in the experimental data between groups. As shown in Table 2, the maximum absolute error of the 10 experimental measurement data was 0.2365 μm.

### 3.2. Angular Velocity Measurement

Figure 6 shows the probability density distribution histogram of 500 sampling points with a rotation speed of 0.5 rpm for 10 repeated records. According to the histogram, more than 95% of the 5000 angular velocity data were 0.50 ± 0.03 rpm. According to Table 3, the average value of the angular velocity data measured in each experiment was 0.50 ± 0.015 rpm, and the maximum error was 0.04305 rpm. According to the above results, when the device was at the lowest angular velocity, there was no gross error in the 5000 angular velocity data, which basically conformed to the error accuracy range.

### 3.3. Optimization of Cell Rotation Speed

In order to find the optimal rotation speed, we carried out a rotation culture at different speeds on the cells before pulling the culture. Before the culture, the number of cells in each sector was the largest and evenly distributed on the circumference of the membrane (Figure 7A). After 5 days of static culture, the total number of cells decreased, although the culture pedestal did not rotate (Figure 7B). It can be seen from Figure 8 that due to the gravity of the cells, the cells attached to the side or below were easy to detach from the membrane. The 120–240° sector had the most detached cells, the 60–120° and 240–300° sectors had less detached cells, and the 0–60° and 300–360° sectors had the least detached cells. When the rotation speed of the cell culture seat was 5 r/d, after 5 days of rotation culture at this speed, compared with the static (0 r/d) culture, the number of cells in the 120–240° sector was significantly increased, and the distribution density of cells in each sector became uniform. Under the influence of rotational speed, cells begin to be uniformly distributed on the whole circumference, but the rotational speed was slow, and the influence of gravity was large at this time, so the overall number of cells decreased (Figure 7C). As shown in Figure 7D, the rotation speed of the cell culture base was 10 r/d, and after 5 days of rotation culture at this speed, the number of cells in the 120–240° sector was significantly increased, and the overall number of cells was also significantly increased compared with the rotation culture at the speed of 5 r/d. The distribution density of cells in each sector was close, and they were uniformly attached to the circumference of the cell culture membrane. When the cells were about to fall off due to gravity, they would rotate the culture seat at a certain speed to make the attached cells turn to the upper side, so as to be affected by the supporting force without immediately leaving the culture seat. Over time, the cells would become completely attached to the culture seat. As shown in Figure 7E, the rotation speed of the cell culture seat was 15 r/d. After rotating and culturing at this speed for 5 days, it was found that with the increase of the rotation speed, the statistical number of cells in each sector decreased, and the total number of somatic cells was less than the total number of cells before the culture. As shown in Figure 7F, the rotation speed of the cell culture seat was 20 r/d. After 5 days of rotating and culturing at this speed, the centrifugal force was far greater than the effect of gravity due to the excessive rotating speed, which made most of the cells attached to the culture base fall off, and the number of cells in each sector decreased significantly.

As shown in Figure 8, the number of cells before the static culture was normalized to 100%. When the angular velocity was 0 r/d, the number of cells in the 120–240° sector was the least, the number of cells in the 0–60° and 300–360° sectors was the most, and the number of cells in the 60–120° and 240–300° sectors was next. The number of cells in each sector was significantly different. As the angular velocity gradually increased, the number of cells in each sector also showed a gradual increasing trend. When the angular velocity was 15 r/d, the angular velocity was inversely proportional to the number of cells. At 20 r/d, the number of cells in each sector decreased significantly. When the angular velocity was 10 r/d, the number of cells in each sector was evenly distributed. We measured a 15% decrease between the number of cells after 5 days and the one at the beginning of the experiment. Therefore, when the angular velocity of the cell culture base was 10 r/d, it could not only ensure that the cells adhered to the wall evenly, but also ensured that the number of cells was sufficient.

### 3.4. Neural Cells Stretch Growth

After 10 days of rotation culture with an angular velocity of 10 r/d, the neuron axons spanned the dividing region between the gap of the sheared film. The stepper motor then began to move at a step pitch of 1 μm per step, giving the cells a rest time of 86.4 s per step. In the process of pulling the culture, the film still maintained a slow rotation speed of 10 revolutions per day, so that the cells could be more fully in contact with the culture medium. After 5 days of the horizontal pulling culture, the axons were not broken. Then, we stained the derived neurons by immunofluorescence to see axons and cell nuclei (Figure 9).

## 4. Discussion

In this study, based on the principle of two-dimensional mechanical force regulating axon growth, a three-dimensional nerve stretching growth device for an implantable nerve interface was developed. The device could realize rotation and straight-line movement. The circular basal culture base was used as the cell carrier, and the uniform rotation of the culture base ensured that the cells adhered to the films uniformly, and the straight-line movement realized the axon stretching. We carried out a displacement accuracy test and angular velocity accuracy test on the device respectively. More than 95% of the data measured by the device under the single step test of 1 μm were 1.0012 ± 0.166 μm, which basically met the error accuracy requirements, and more than 90% of the relative errors were less than 2%. In the test data of the minimum speed of 0.5 rpm, more than 95% of the data were 0.50 ± 0.03 rpm, and there was no gross error, which met the accuracy requirements of the experiment. The results of two test experiments showed that the three-dimensional axon stretching growth device can be used for an axon-stretching growth culture of nerve cells.

In order to explore the effect of different rotating speeds of the circular substrate on the number of cells attached to the culture base. We conducted corresponding experiments for different rotating speeds (Table 1) and completed a statistical analysis of cell numbers. The number of cells in each sector was extremely uneven when cultured in the static or at an angular rate of 5 r/d. When cultured at the angular velocity of 15 r/d and 20 r/d, although the cells in each sector were evenly distributed, the total number of cells was significantly reduced. The number of cells would be further reduced after the pulling culture [19]. When the cell culture stand was rotated at an angular speed of 10 r/d, the cells in each sector were evenly distributed, and the total number of cells was 85% of the total number of cells before culture. As the rotation speed increased, the cells were more evenly distributed around the circumference. However, when the rotation speed reached 20 r/d, the cells were evenly distributed on the whole circumference, but at this rotation speed, the total number of cells was less than 50% of the number of cells before the culture (Figure 7 and Figure 8). Therefore, we should not only ensure that the cells are evenly distributed in the circumference, but also maintain that the total number of cells is more than 80% of that before the experiment. Finally, through the comparison of experimental data under different rotational speeds, we found that when the basal rotation speed was 10 r/d, it met our expectations, which could not only ensure the uniform adherence of cells, but also ensure the sufficient number of cells.

As we all know, stem cell technology expands our expectation for function reconstruction. In this experiment, we confirmed the feasibility of stem cells in biological engineering by combing with a three-dimensional stretch growth model. The derived cells were successfully pulled in this device. However, whether the physiological indexes of the stretch growth neurons are normal and whether they can accurately transmit nerve signals needs further verification. In recent years, Chen et al. [20,21] induced human embryonic stem cells into cerebral cortical neurons and obtained a nerve bundle with a length of about 1 cm by pulling and culturing integrated neurons. Through the combination of patch clamp and photo gene technology, it was found that the stretch growth nerve axons could still conduct action potentials. However, they used a two-dimensional method to pull and cultivate nerve cells.

The nerve bundle cultured by two-dimensional mechanical traction had a complete cytoskeleton structure [22] and could conduct nerve signals [23], which not only shows a very good application prospect in the repair of nerve injury, but also provides a new idea for the development of neural interface [9]. However, at present, the laboratory generally uses the adherent method to culture nerve cells, resulting in the cells generally in a two-dimensional state. However, the nerve bundle in the biological body is a three-dimensional cylinder, and only the three-dimensional shape of the nerve bundle can be more directly applied to patients. Secondly, peripheral nerves are generally layered, and cultured monolayer nerve cells cannot replace autologous nerve tissue. In our cylindrical traction culture experiments, although the cell bodies were in an adherent culture, the intermediate axons were suspended in the culture medium. When the suspended axons were further cultured, they aggregated into fascicles [19]. In addition, in the future, we will further study and design a new multi-layer structure (Figure 10) based on this structure. This allows us to design the arrangement of our nerve bundles according to the cross-section of the amputee and even to stratify the sensory, motor, and glial cells. The implant formed in this way can be directly transplanted, which greatly improves the efficiency of the neural interface prosthesis installation.

There are still many shortcomings in this experimental device. For the motion with a single step of 1 μm, we chose to use a stepping motor, which was an open-loop control mode. It is not clear whether the distribution and number of cells will have better results at other finer rotational speeds. In addition, the device needs to be placed in the cell incubator to maintain the environment required for cell growth, which is inconvenient for experimental observation and easy to cause axon fracture during observation and replacement of culture medium. In the future, we can develop a brand-new experimental device, which can realize closed-loop control, easy observation and speed regulation, and simple experimental operation, and the device itself can maintain the environment required for culturing cells without putting it into the incubator [24].

## 5. Conclusions

In this study, according to the principle of mechanical force regulating axon growth, a three-dimensional axon stretching culture system for an implantable neural interface was developed. The system could realize rotation and straight-line movement. The circular basal culture base was used as the axon carrier. The uniform rotation of the culture base ensured that the nerve cells adhered to the film evenly, and the straight-line movement realized the stretching of the nerve axons. STM32F407 was used as the main control chip to coordinate and control three motors to make axons grow in a given direction and speed and maintain nutrition supply. Finally, the displacement accuracy test and angular velocity accuracy test were carried out on the device, and the adherence of cells at different rotational speeds was observed. The statistical distribution of cells was analyzed to verify the stability and reliability of the experimental device. The experimental results showed that there was no gross error in the displacement accuracy of 1μm and the rotation speed accuracy of 0.5 rpm of the system. In addition, we could clearly see axons and nuclei of the cells cultured by rotating and stretching. In the future, the neural tissue with a circuit interface cultivated by the device can be used for implantable neural interfaces to promote the recovery of sensory and motor functions of patients.

## Figures and Tables

**Figure 1 micromachines-13-01558-f001:**
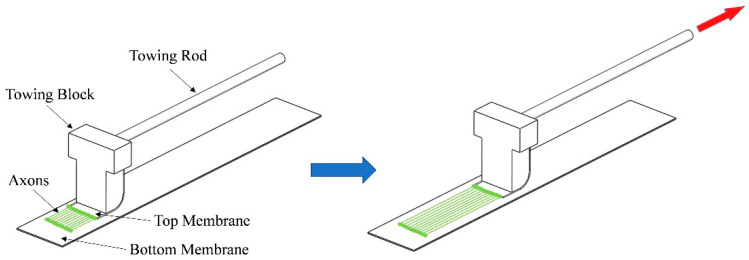
The working principle of two-dimensional axon stretch growth.

**Figure 2 micromachines-13-01558-f002:**
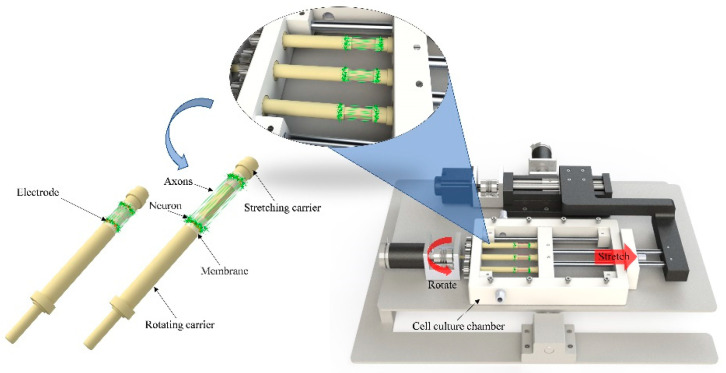
The working principle of the three-dimensional neural cell stretch growth device.

**Figure 3 micromachines-13-01558-f003:**
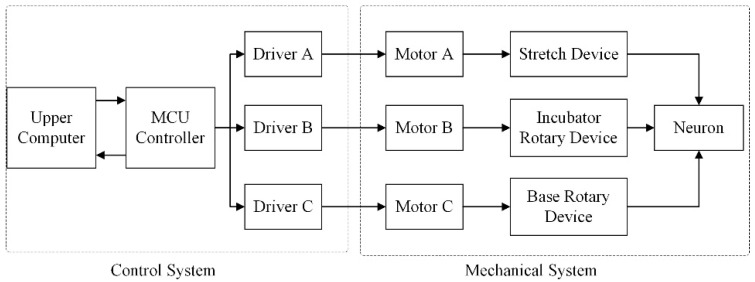
The control circuit of the cell stretch growth device.

**Figure 4 micromachines-13-01558-f004:**
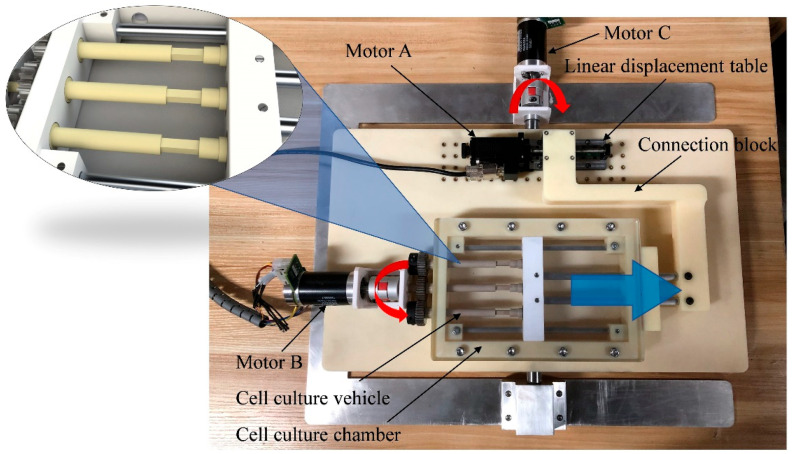
The three-dimensional nerve stretch growth device.

**Figure 5 micromachines-13-01558-f005:**
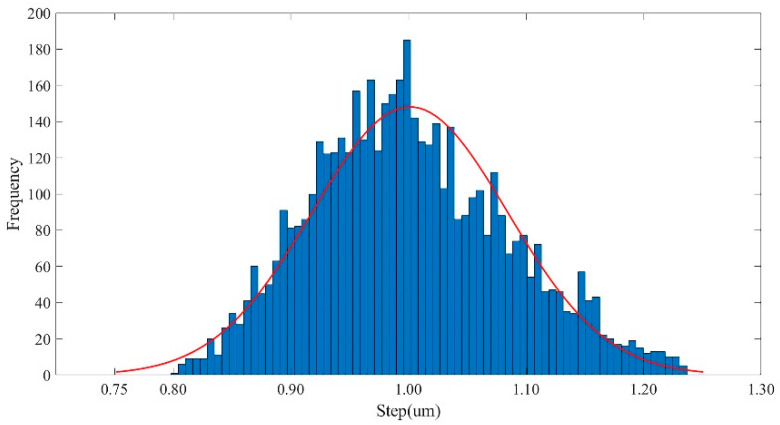
The probability density distribution histogram of the neuron stretch device with a one-μm step.

**Figure 6 micromachines-13-01558-f006:**
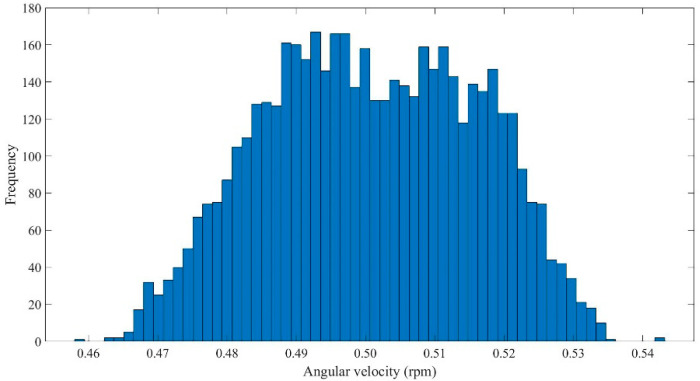
The probability density distribution histogram of the neuron stretch device with 0.5 rpm rotation speed.

**Figure 7 micromachines-13-01558-f007:**
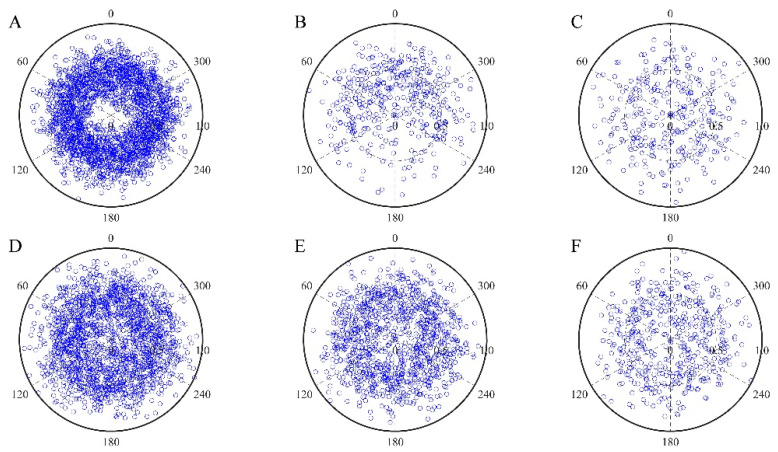
Polar plots of cell distribution. Each blue circle represents a cell. The values of 0, 60, 120, 180, 240, and 300 represent the angles starting counterclockwise from directly above the annular film. The value of 0.5 represents the slit where the film was cut, and almost all cells were placed within 5 mm around the slit; 5 mm to the left represents 0; and 5 mm to the right represents 1. (**A**): The number of cells before the static culture after the cells were placed around the cell culture membrane. (**B**): The number of cells in each sector after the static culture for 5 days at a rotation speed of 0 r/d. (**C**): The number of cells in each sector after the static culture for 5 days at a speed of 5 r/d. (**D**): The number of cells in each sector after the static culture for 5 days at a speed of 10 r/d. (**E**): The number of cells in each sector after the static culture for 5 days at a speed of 15 r/d. (**F**): The number of cells in each sector after the static culture for 5 days at a speed of 20 r/d.

**Figure 8 micromachines-13-01558-f008:**
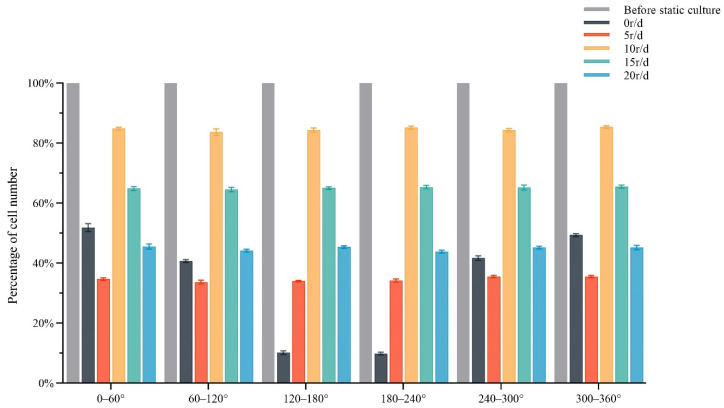
The bar chart of cell distribution.

**Figure 9 micromachines-13-01558-f009:**
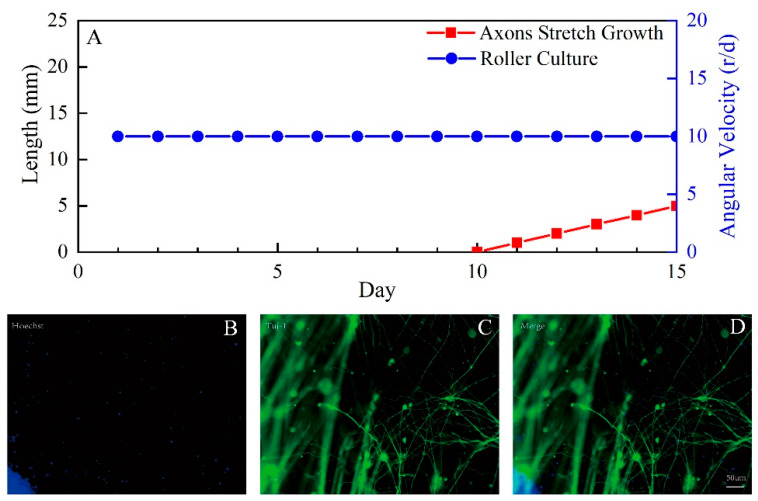
Immunofluorescence analysis of axons. (**A**) Flow Diagram of Experiment. (**B**) Antibodies against Hoechst (blue) showing the cell nucleus. (**C**) Antibodies against TuJ-1 (green) showing axons. (**D**) Merge. Scale bars: 50 μm.

**Figure 10 micromachines-13-01558-f010:**
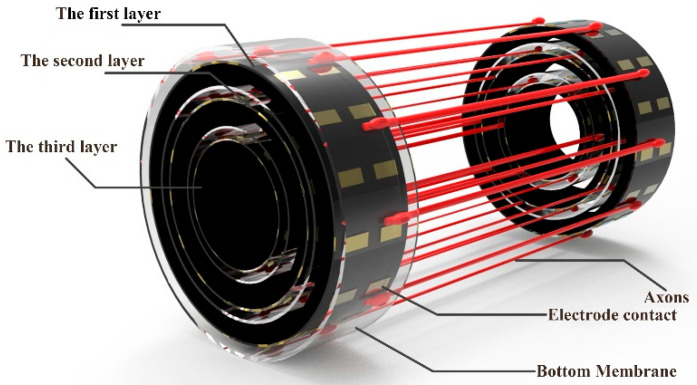
Design of a multi-layer three-dimensional nerve traction culture device.

**Table 1 micromachines-13-01558-t001:** Cell exfoliation test.

Type	Rotational Speed (r/d)	Time(d)
Rotate	0	5
Rotate	5	5
Rotate	10	5
Rotate	15	5
Rotate	20	5

**Table 2 micromachines-13-01558-t002:** The statistical analysis of neuron stretch device with a one-μm step.

Number	1	2	3	4	5	6	7	8	9	10
Average value	1.0022	1.0026	1.0002	1.0022	0.9996	1.0052	0.9969	0.9965	1.0044	1.0023
Variance	0.0071	0.0067	0.0064	0.0066	0.0072	0.0076	0.0071	0.0070	0.0069	0.0068
Maximum error	0.2365	0.2250	0.2170	0.2170	0.2070	0.2320	0.2315	0.2330	0.2300	0.2250

**Table 3 micromachines-13-01558-t003:** The statistical analysis of the neuron stretch device with 0.5 rpm rotation speed.

Number	1	2	3	4	5	6	7	8	9	10
Average value	0.500824	0.500669	0.500655	0.500598	0.500863	0.500707	0.500497	0.500937	0.500631	0.500847
Variance	0.000258	0.000221	0.000223	0.000222	0.000225	0.000234	0.00023	0.000235	0.000233	0.000231
Maximum error	0.04195	0.03405	0.0335	0.03385	0.0352	0.04305	0.03675	0.03415	0.04255	0.03575

## Data Availability

Data presented in this study are available in this article.

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
