# Peer review of "Development of a Three-Dimensional Nerve Stretch Growth Device towards an Implantable Neural Interface"

_micromachines, 2022, doi:10.3390/mi13101558_

Round 1
Reviewer 1 Report
The study entitled "Development of three-dimensional nerve stretch growth device for implantable neural interface" by Xiao Li and co-workers developed a three-dimensional nerve stretch growth device for implantable neural inter-face. A circular basal culture base was used as the axon carrier. The culture system could realize rotation and straight-line movement which ensured the firm cell attachment and the stretching of the nerve axons. The displacement accuracy test and angular velocity accuracy test were carried out on the culture device, and the adherence of cells at different rotational speeds was observed.
However, the main concern is the study present in the paper is not a true 3D model. In my understanding, the whole work described in this paper is a 2D neuron culture in a dynamic culture system. Previous studies have been carried out under two-dimensional culture system which is also described as Figure 1 in the paper. Therefore, the study is lack of the novelty and significance.
My overall decision is to reject this work to be published under Micromachines.
Reviewer 2 Report
This manuscript puts forward an innovative three-dimensional structure design scheme for implantable neural interface. The author designed a multi-motor driven equipment that can stretch and rotate nerve cells, and provided a design method of three-dimensional nerve interface. However, the following concerns need to be carefully taken before further desicion.
(1) Page 2 Line 94: “...leave an overlapping area of 3-5 mm between the two membranes.” If the “overlapping area of 3-5 mm” means the width of this area is 3-5 mm? The sentence should be rephrased to avoid misunderstanding.
(2) Page 3 Line 98: the sentence ” so as to pull the nerve axons across the membrane” can be revised as “ so as to pull the nerve axon toward the top membrane” which can be clearer to readers.
(3) Page 8 discussed the influence of different cell rotation speed. Will other factors affect cell culture such as the diameter of towing rod?
(4) In this manuscript, only the model diagram and the physical diagram of the device are shown, but the pictures of the test process are not intuitive enough.
(5) Figure 10. The Structure of culture base, it can be merged with Figure 2 or Figure 4.
Reviewer 3 Report
In the paper “Development of three-dimensional nerve stretch growth device 2 for implantable neural interface”, Xiao Li and colleagues describe the realization of a device that can allow horizontal nerve stretch growth and torsion growth simultaneously, thanks to the use of an arrangement of linear and stepper motors synchronously controlled. They performed a series of experiments to quantify the device precision both in linear and rotation movements. They describe the effect of the rotation on the number of cells that remain attached to the device under different rotating speed and they describe also a proof-of-principle neural cell stretch growth application. The paper itself is well written and easy to read, the level of scientific findings is unfortunately limited, and the main claims are not totally clear. For these reasons I am afraid I cannot recommend the paper for publication in its current version, but in the following I suggest some points on which the paper should be improved. Proven that these points are addressed by the authors, I will be happy to reconsider my current opinion.
1. In the introduction, only a final sentence is dedicated to describe the findings of the study, accounting for less than the 5% of the total length. It would be beneficial for the impact of the study to outline the main findings in a more detailed fashion in the introduction.
2. The materials and methods section opens with a paragraph on two-dimensional axon stretch configurations, but the topic of the paper is on three-dimensional growth. This description could go in the introduction and it is not relevant enough in the study to have a corresponding figure assigned.
3. The authors should standardize the measurement units across the entire paper. Example: the Greek letter µ is often written as “u” (i.e., um) “µ” (µm), “µ” but using a different font, and even “micron” in some points. Standardize also the way the units follow the numerical values: XXµm or XX[space]µm.
4. In the last paragraph of section 2.3 it is not clear how the angular torsion is performed. How the different values of round per day are reached? By shortening the interval between two consecutive steps or by lengthen the single step? The motor speed was set at 0.5 rpm, but this would mean a way bigger number of rounds per day. These points should be clarified for sake of understanding by a future reader.
5. Figure 5 is referred to as a probability density distribution histogram, while it is a frequency histogram, as written in the figure caption.
6. Are the measurements performed in 3.1 repeated on the same system under the same conditions? If so, the average values can be given in mean plus minus standard deviation instead of as an interval. Also because at the end a total average value is given for the mean and the variance. Because a fit is shown in the figure, usual statistical quantities could be provided (µ,σ, a goodness of fit evaluator).
7. There are two Table 2, one for Section 3.1 and one for section 3.2. Please address the typo.
8. Why the authors say “Moreover, the higher the speed of the servo motor, the higher the accuracy (data not shown)”? Is it an important point for the discussion of the results? If not, it can be removed as no data are shown in support of the sentence.
9. Same observations for the previous section apply also for this one (points 4 and 5 of this report). In addition, it clearly seems that the data in Figure 6 do not follow a normal distribution. On the other hand, I do not think that it is important for these data to be distributed in that way, as they are basically setup calibration data. The authors should discuss about the possibility of removing it, if they consider it the case.
10. Lines 277-278 “After rotating and culturing at this speed for 5 days, it is found that with the increase of the rotation speed, the higher the speed, the greater the centrifugal force.” This is trivial and it can be removed. Discussions for Figure 7E (15r/d) and 7F (20r/d) say the same thing, but slightly rephrased. They can be unified in a single discussion.
11. The observation “centrifugal force was greater than the effect of gravity” is referred only at the particular measurement or in general? Because from figure 8 it seems that when the rotation is slow, hence when the effect of gravity is prominent, the number of cells decreases strongly over the 5 days time span compared to higher speeds. Would it be correct to say that with a speed of 10 r/d the effect of gravity on the variation of the number of cells is minimized but for higher speeds the number of cells lost by centrifugal forces effect is stronger?
12. In Figure 8 also the data for the cell number before culture (the ones in Figure 7) should be added, to allow for a straightforward comparison. Avoid expressions like “the difference between the total number of cells and the number of cells at the beginning of the experiment was very small”, instead quantify the reduction (in this example) as “We measure a XX% decreasing between the number of cells after 5 days and the one at the beginning of the experiment”. This is partially done in the discussion section (section 4) but it should be addressed over the entire paper.
13. How the error bars in figure 8 are calculated? Was the experiment performed one time per speed or was it repeated more than one time to obtain an error bar?
14. Figure 9B-D are barely commented in the main text Are the images taken before or after the stretch, are them taken on a totally different culture? Which is the message coming out from these images?
15. The title “Development of three-dimensional nerve stretch growth device for implantable neural interface” is a bit misleading, as no further verification is performed on the cells after stretching to show the electrical signal transmission or cell functionality. Neural stretch culture has to be integrated with other components to realize a neural interface, and the presented work does not address the “neural interfacing” related problems. I suggest removing the expression “neural interfaces” in the title or to reduce the overall claim suggesting the possibility to use the technology in neural interfaces with something like “Development of three-dimensional nerve stretch growth device TOWARDS implantable neural interface”.
16. Why is Figure 10 placed in the discussion section? Won’t it be better suited for the materials and methods section? Also, the description of this figure (lines 356-367) could be rewritten for a better understanding.
17. Overall, it is not totally clear which is the clear advantage of 3D growing over 2D growing and at which extent the technology and findings reported in this paper are new or incremental with respect to literature. How the technology here described can improve the cell culturing? Which advantages could this bring in a future neural interface? These points should be discussed in the conclusions.
Round 2
Reviewer 1 Report
The 3D culture is well defined actually. As the author answered, the paper presents the culture of cells on a cylinder. Without any images/immunostainings as a proof of "3D" culture, I will still give my recommendation as rejection.
Reviewer 3 Report
Dear authors,
thank you for the effort you made in addressing all my points. I believe that the discussion of the results has improved enough since the first submitted version. For this reason, I can recommend the paper for publication in its revised form.